# Intestinal Morphology and Glucose Transporter Gene Expression under a Chronic Intake of High Sucrose

**DOI:** 10.3390/nu16020196

**Published:** 2024-01-07

**Authors:** Kana Yamamoto, Norio Harada, Takuma Yasuda, Tomonobu Hatoko, Naoki Wada, Xuejing Lu, Yohei Seno, Takashi Kurihara, Shunsuke Yamane, Nobuya Inagaki

**Affiliations:** 1Department of Diabetes, Endocrinology and Nutrition, Graduate School of Medicine, Kyoto University, Kyoto 606-8507, Japan; 2P.I.I.F. Tazuke-Kofukai Medical Research Institute, Kitano Hospital, Osaka 530-8480, Japan

**Keywords:** high-sucrose diet, fructose, intestinal morphology, glucose transporter, α-glucosidase inhibitor

## Abstract

Sucrose is a disaccharide that is degraded into fructose and glucose in the small intestine. High-sucrose and high-fructose diets have been reported, using two-dimensional imaging, to alter the intestinal morphology and the expression of genes associated with sugar transport, such as sodium glucose co-transporter 1 (SGLT1), glucose transporter 2 (GLUT2), and glucose transporter 5 (GLUT5). However, it remains unclear how high-fructose and high-sucrose diets affect the expression of sugar transporters and the intestinal morphology in the whole intestine. We investigate the influence of a chronic high-sucrose diet on the expression of the genes associated with sugar transport as well as its effects on the intestinal morphology using 3D imaging. High sucrose was found to increase GLUT2 and GLUT5 mRNA levels without significant changes in the intestinal morphology using 3D imaging. On the other hand, the delay in sucrose absorption by an α-glucosidase inhibitor significantly improved the intestinal morphology and the expression levels of SGLT1, GLUT2, and GLUT5 mRNA in the distal small intestine to levels similar to those in the proximal small intestine, thereby improving glycemic control after both glucose and sucrose loading. These results reveal the effects of chronic high-sugar exposure on glucose absorption and changes in the intestinal morphology.

## 1. Introduction

The excessive intake of sucrose is a worldwide health concern and is associated with a high risk of type 2 diabetes, cardiovascular disease, and all-cause mortality [1,2,3]. The consumption of added sugars, including sucrose, remains above the recommended levels, despite the gradual decrease since 2004 [4,5,6]. Several reports have found that an excessive intake of added sugars induces insulin resistance, fatty liver, and dyslipidemia by stimulating de novo lipogenesis and the inhibition of fatty acid oxidation [7,8].

Carbohydrates and some sugars are degraded into monosaccharides by intestinal enzymes; these monosaccharides are absorbed by the villi of the intestine through various glucose transporters. Thus, the intestinal morphology and glucose transporter gene expression in the intestine are crucial as the first step of sugar metabolism in vivo. Sucrose is a disaccharide; it is degraded into fructose and glucose by intestinal sucrase [9]. Fructose and glucose are absorbed by the intestinal epithelial cells of the proximal small intestine; the absorbed glucose is then transported from the intestinal epithelial cells to the blood vessels. On the other hand, fructose is partially converted into glucose by ketohexokinase in the intestinal epithelial cells [10]. Glucose is then transported to the blood vessels.

Previous investigations of the intestinal morphology by two-dimensional (2D) imaging using histology have reported that the long-term intake of sucrose and fructose elongates the villi in the duodenum and jejunum [11,12]. However, the analysis of the intestinal morphology by tissue sectioning is limited to the small intestine; we recently established a method to evaluate the intestinal morphology using tissue clearing and three-dimensional (3D) imaging [13].

Sodium glucose co-transporter 1 (SGLT1) is localized on the apical side of enterocytes; glucose transporter 2 (GLUT2) and glucose transporter 5 (GLUT5) are localized on the basolateral as well as the apical sides of enterocytes [14,15,16]. SGLT1 participates in glucose absorption; GLUT2 participates in glucose and fructose absorption; and GLUT5 participates in fructose absorption [14,15,16]. It has been reported that the short-term intake of a high-fructose diet increases the expression levels of SGLT1, GLUT2, and GLUT5 mRNA [17]. However, it is unclear how the long-term intake of high-fructose and high-sucrose diets affects the expression of glucose transporters in the whole intestine.

In this study, we comprehensively investigate the influence of the chronic intake of a high-sucrose diet on the expression of the genes associated with sugar transport as well as the intestinal morphology using 3D imaging in the whole intestine. We also evaluate the effects of a chronic high-sucrose intake on blood glucose levels after glucose and fructose ingestion.

## 2. Materials and Methods

### 2.1. Experimental Animals and Diets

C57BL/6-background wild-type (WT) male mice and Villin-1 reporter male mice were used for the analysis of gene expression in the intestine and the intestinal morphology, respectively. All mice were housed under a 14 h/10 h light/dark cycle with free access to food and water. Animal care and procedures were approved by the Kyoto University Animal Care Committee (No. Med Kyo20217, approval date: 25 March 2020).

The α-glucosidase inhibitor (αGI) miglitol was used to delay sucrose absorption in the small intestine by inhibiting the degradation of sucrose [18,19,20]. The effective percentage of miglitol was determined by referring to previous studies [21,22]. Six-week-old WT and Villin-1 reporter mice were divided into four feeding groups: control diet (CD) with an energy density of 3.43 kcal/g (CLEA rodent diet CE-2: 58.0% carbohydrate, 29.2% protein, and 12.8% fat consisting of soy oil, with the carbohydrates consisting of starch; CLEA Japan, Tokyo, Japan); high-sucrose diet (SD) with an energy density of 3.76 kcal/g 37.3% sucrose (Appendix A); CD containing 0.08% miglitol (CD+αGI); and SD containing 0.08% miglitol (SD+αGI), all for 10 weeks.

For the investigation of the gene expression in the WT mice, food intake for 24 h was measured after 9 weeks of diet feeding. After 10, 11, and 13 weeks of diet feeding, insulin tolerance tests (ITTs), oral glucose tolerance tests (OGTTs), and oral fructose tolerance tests (OFTTs) were performed. The mice were then sacrificed, and the lengths of the small intestine and colon and the weights of the inguinal white adipose tissue (iWAT), epididymal WAT (eWAT), perirenal WAT (pWAT), liver, and gastrocnemius muscle to represent the skeletal muscle were measured [23].

Villin-1 encodes the actin-binding protein Villin, which is expressed in intestinal epithelial cells [13,24,25]. Villin-1 reporter mice were generated by crossbreeding Villin-1-Cre transgenic mice and Ai14 homozygous mice (JAX stock #004586, #007908) (Jackson Laboratory, Bar Harbor, ME, USA). These mice enabled the visualization of intestinal epithelial cells by tdTomato, a fluorescent protein. Intestinal morphology was analyzed using the reporter mice after 13 weeks of diet feeding.

### 2.2. ITT, OGTT, and OFTT

Soon after the 10-week duration of the diet period, the mice were housed in cages without food, to match the fasting condition of the WT mice. ITTs were performed by the intraperitoneal injection of human regular insulin (100 U/mL, Eli Lilly and Company, Indianapolis, IN, USA) after fasting for 4 h [25,26]. Blood samples were taken at 0, 30, 60, 90, and 120 min from the tail vein after insulin administration. Glucose levels were analyzed by the percent change from blood glucose levels at 0 min.

OGTTs and OFTTs were performed after fasting for 16 h. Blood samples were collected at 0, 15, 30, 60, and 120 min from the tail vein after the oral administration of glucose and fructose (2 g/kg body weight) [25,27]. Blood glucose levels were measured by the glucose oxidase method (Sanwa Kagaku Kenkyusho Co., Ltd., Nagoya, Japan). Plasma insulin levels were measured using a mouse insulin enzyme-linked immunosorbent assay (ELISA) kit (FUJIFILM Wako Shibayagi Corporation, Shibukawa, Japan).

### 2.3. Quantitative Reverse-Transcription Polymerase Chain Reaction (RT-PCR)

Five equally spaced sections of 1 cm of small intestine were collected from the oral side (S1) to the anal side (S5) (Appendix A). Total RNAs of the small intestine were extracted with TRIzol reagent (Thermo Fisher Scientific, Waltham, MA, USA) [25]. Total RNAs were reverse transcribed using PrimerScript RT reagent Kit with gDNA Eraser (Takara Bio Inc., Kusatsu, Japan); mRNA expression levels were quantified by using cDNA dilution curves in real-time PCR with the ABI PRISM 7000 Sequence Detection System (Applied Biosystems Inc., Waltham, MA, USA). As housekeeping genes, the expression levels of the PPIA gene were assessed. The sequences of the primers are listed in Appendix A.

### 2.4. Collection of Intestine Samples and Tissue Clearing

Tissue sections were collected from 12 a.m. to 4 p.m. on the day following the OFTTs. Villin-1 reporter mice were anesthetized, followed by perfusion with phosphate-buffered saline (PBS) and 4% paraformaldehyde (PFA) (Wako Pure Chemical Industries, Osaka, Japan) [28]. Five sections of 1 cm of small intestine from the oral side (S1) to the anal side (S5) and three sections of colon from the oral side (C1) to the anal side (C3) were collected (Appendix A). Tissue clearing was performed as previously described [13].

### 2.5. Image Acquisition and Processing

The 3D fluorescence images were acquired by spinning disk confocal microscopy (Dragonfly, Andor Technology Ltd., Belfast, UK) on an ECLIPSE Ti2-E (Nikon Solutions Co., Ltd., Tokyo, Japan) device through a PLAN APO 20× objective lens (Nikon, numerical aperture [NA], 0.8), followed by analysis with Imaris Version 9.5.0 (Bitplane AG, Zurich, Switzerland) as previously described [13,29]. Villus length, width of the villus major and minor axes, crypt depth, and width of crypt axis were then measured (Appendix A). Averages of the parameters of ten villi or crypts in each mouse were calculated and analyzed as the values of one sample.

### 2.6. Statistical an Alysis

The results are shown as a dot plot or the mean ± SEM. One or two sample data that were outside of the mean ± 2 SD were excluded. Statistical significance was determined by Student’s *t*-test for comparison between the two groups CD and CD+αGI; and ANOVA with Tukey test for comparison among the three groups CD, SD, and SD+αGI, using JMP Pro statistical software (version 16.2.0) (SAS Institute, Cary, NC, USA). *p*-values below 0.05 were regarded as statistically significant.

## 3. Results

### 3.1. Effect of Sucrose Diet on Body and Tissue Weight and Insulin Sensitivity

After 10 weeks of diet feeding, the body weight of the SD group was not significantly different from that of the CD group, while that of SD+αGI group was significantly less than that of CD mice (Figure 1A). In the ITTs, the reduction in blood glucose was greater in SD and SD+αGI than that in CD mice, indicating that SD and SD+αGI mice have greater insulin sensitivity than CD mice (Figure 1B). The weight of iWAT, eWAT, pWAT, and skeletal muscle was not significantly different among the three groups, whereas SD+αGI mice showed a reduced liver weight compared with CD mice (Figure 1C–E). Food intake did not differ among the three groups (Figure 1F).

The body weight of CD+αGI mice was not significantly different from that of CD mice, except that at 2 and 3 weeks after the initiation of diet feeding (Appendix A). There was no difference in blood glucose levels by ITT between the two groups (Appendix A). Food intake and the weight of each tissue did not differ between the two groups (Appendix A).

### 3.2. Morphology of Small Intestine

The length of the small intestine of SD mice was less than that of CD mice, but there was no significant difference in this parameter between CD and SD+αGI mice (Figure 2A). While the villus length in S1 was not significantly different among the three groups, the villus length in S2 of SD and SD+αGI mice was greater than that of CD mice (Figure 2B). Furthermore, the villus length in S3 and S4 of SD+αGI mice was significantly greater than that of CD and SD mice. The width of the villus major axis in S4 and S5 of SD+αGI mice was greater than that of SD mice (Figure 2C,E). On the other hand, the width of the villus minor axis was not significantly different in any section of the small intestine (Figure 2D,E). Crypt depth and width of crypt axis did not significantly differ among the three groups, except those in S5 of SD+αGI mice, which were greater than those in S5 of SD mice (Figure 2F–H).

The length of the small intestine of CD+αGI mice was not significantly different from that of CD mice (Appendix A). The villus length in S3 and S4 and the width of the major axis in S3 of CD+αGI mice were greater than those of CD mice, while the other parameters did not differ between the two groups (Appendix A).

### 3.3. Morphology of Colon

The length of the colon, crypt depth, and width of the crypt axis did not significantly differ among the three groups (Figure 3A–D). These parameters in CD and CD+αGI mice did not differ (Appendix A). These results indicate that the chronic intake of high sucrose and αGI treatment do not affect colon morphology.

### 3.4. Gene Expression of mRNA Involved in Glucose and Fructose Absorption in Small Intestine

The expression levels of SGLT1 mRNA were not significantly different between CD and SD mice (Figure 4A). However, the expression levels of SGLT1 from S3 to S5 were significantly higher in SD+αGI mice than those in CD and SD mice.

The expression levels of GLUT2 mRNA and GLUT5 mRNA in S1 and S2 were dramatically increased in SD compared to those in CD mice (Figure 4B,C). However, these expressions were not significantly different between CD and SD+αGI mice. In S3, S4, and S5, these expression levels were significantly higher in SD+αGI mice compared to those in CD and SD mice.

In CD and SD mice, the expression levels of SGLT1 mRNA, GLUT2 mRNA, and GLUT5 mRNA in S2 were the highest among the sections, whereas those in SD+αGI mice were the highest in S3.

In CD+αGI mice, the expression levels of SGLT1 mRNA were increased from S3 to S5, those of GLUT2 mRNA were increased in S3 and S4, and those of GLUT5 mRNA were increased in S4 compared with those in CD mice (Appendix A).

### 3.5. Blood Glucose and Insulin Levels after Glucose and Fructose Ingestion

During OGTT, the increments in blood glucose levels after glucose loading and the incremental area under the curve of glucose (iAUC-glucose) were significantly higher in SD mice than those in CD mice (Figure 5A). In SD+αGI, the increments in blood glucose levels and iAUC-glucose were significantly lower than those in SD mice. There was no significant difference in the increments in blood glucose levels between CD and SD+αGI mice. The increments in insulin levels did not differ among the three groups (Figure 5B).

During OFTT, the increments in blood glucose levels after fructose loading and iAUC-glucose were significantly higher in SD than those in CD mice (Figure 5C). In SD+αGI mice, the increments in blood glucose levels and iAUC-glucose were significantly lower than those in SD mice. Blood glucose levels did not differ between CD and SD+αGI mice. The increments in insulin levels did not differ among the three groups (Figure 5D).

In CD+αGI mice, the increments in glucose levels at 15 min during OGTT were higher than those in CD mice, while no significant difference was found in iAUC-glucose or the increments in insulin levels (Appendix A). During OFTT, the increments in glucose levels at 120 min in CD+αGI mice were higher than those in CD mice, while iAUC-glucose levels did not differ between the two groups; the increments in insulin levels at 15 min were significantly higher in CD+αGI mice than those in CD mice (Appendix A). The absolute values of blood glucose and insulin levels are shown in Appendix A.

## 4. Discussion

This study comprehensively investigated the effects of a chronic intake of high sucrose by analysis of the gene expressions of glucose transporters and alterations of intestinal morphology associated with sugar transport. We were able to analyze the details of intestinal morphology including the length and width of villi and crypts more precisely by using 3D imaging than by conventional 2D imaging. Regarding the sugar transporters, previous studies analyzed only a part of the proximal small intestine; we evaluated gene expressions associated with sugar transport in the whole intestine.

We find by 3D imaging that intestinal morphology is not altered by the chronic intake of high sucrose except in one section (S2) in the proximal small intestine. Previous 2D studies reported that a high-sucrose-diet intake elongated both the length and width of villi in the duodenum and jejunum, which could be misleading insofar as the data were limited to only a small part of the intestine and were achieved by 2D imaging [11,12]. It has been shown that when intestinal cells become hypoxic, fructose 1-phosphate, a metabolite of fructose, inhibits pyruvate kinase M2 to detoxify reactive oxygen species, which promotes the survival of intestinal epithelial cells and villus length under a high-sucrose diet with high fat [12]. Considering that we did not find a remarkable difference in intestinal morphology between CD and SD mice, the high-sucrose diet with normal fat used in this study might not induce hypoxia in intestinal epithelial cells.

Chronic intake of high sucrose was also found to increase the expression levels of GLUT2 and GLUT5 mRNA in the proximal small intestine, exacerbating glucose intolerance after fructose or glucose intake. Indeed, the expression levels of these sugar transporters are increased with increasing luminal concentrations of glucose and fructose in the small intestine [14,30]. Thus, it is possible that the degradation of excessive sucrose affects the luminal concentration of fructose, which thereby increases the absorption of both glucose and fructose via increased expression levels of sugar transporters in the proximal small intestine. On the other hand, we found that the chronic intake of high sucrose did not increase the expression level of SGLT1 mRNA. However, while glucose uptake via SGLT1 is about twice that via GLUT2 at a high concentration of luminal glucose [31], the increment in glucose absorption via SGLT1 by increased luminal glucose concentration was found to be less than that via GLUT2 and GLUT5 [32,33]. Thus, the alteration of SGLT1 expression by increases in sugar intake might be finely controlled to forestall drastic changes in glycemic control. Additional studies are required to clarify the details of the regulation of the genes associated with sugar transport under a chronic intake of high sucrose.

Our previous study revealed that the cross section of villi in the proximal small intestine is spindle-shaped, while that in the distal small intestine is round-shaped [13]. Interestingly, in this study, the delay in sucrose absorption caused by αGI under a chronic high sucrose intake was found to change the cross section of villi in the distal small intestine from round-shaped to spindle-shaped, as shown in Figure 2E. In addition, the delay in sucrose absorption reduced the expression levels of GLUT2 and GLUT5 mRNA, but not that of SGLT1 mRNA, in the proximal small intestine; it also increased the expression levels of GLUT2, GLUT5, and SGLT1 mRNA in the distal small intestine. In addition, it has been reported that the expression levels of sugar transporters are elevated by high luminal concentrations of glucose and fructose in the small intestine [14]. Thus, our results indicate that the delay in sucrose absorption increases the luminal concentrations of glucose and fructose in the distal small intestine, thereby altering intestinal morphology as well as the polarity of sugar transporter expression in the distal small intestine to resemble that in the proximal small intestine. Thus, such an alteration of the intestinal sections where sugar is mainly absorbed might well lead to a reduction in blood glucose levels after excessive glucose or fructose loading to levels that are similar to those under control diet feeding.

While treatment with αGI under control diet feeding increased the expression levels of SGLT1 and GLUT2 mRNA in the distal small intestine, the changes in the expression levels of GLUT5 mRNA and intestinal morphology were relatively small. Moreover, treatment with αGI did not change blood glucose levels or AUC-glucose after glucose or fructose loading, except for a single change at 15 min after glucose loading. The control diet consisted mainly of carbohydrates, which are polysaccharides that are degraded more slowly than disaccharides [34]. In addition, the control diet did not contain fructose, which is transported in intestinal epithelial cells via GLUT5. Therefore, the effect of delayed sugar absorption from αGI on intestinal morphology, the gene expression of glucose transporters in the small intestine, and postprandial blood glucose levels likely depend on the form and type of sugar.

## 5. Conclusions

This study shows by 3D imaging that a chronic intake of high sucrose increases gene expression levels associated with sugar transport, without any remarkable changes in intestinal morphology. Moreover, the delay in sucrose absorption under a chronic high sucrose intake alters intestinal morphology and the polarity of gene expression associated with sugar transport, thus contributing to an improvement in glycemic control after glucose or fructose loading. These results provide novel insight into the effects of a chronic intake of high sucrose and changes in sugar absorption.

## Figures and Tables

**Figure 1 nutrients-16-00196-f001:**
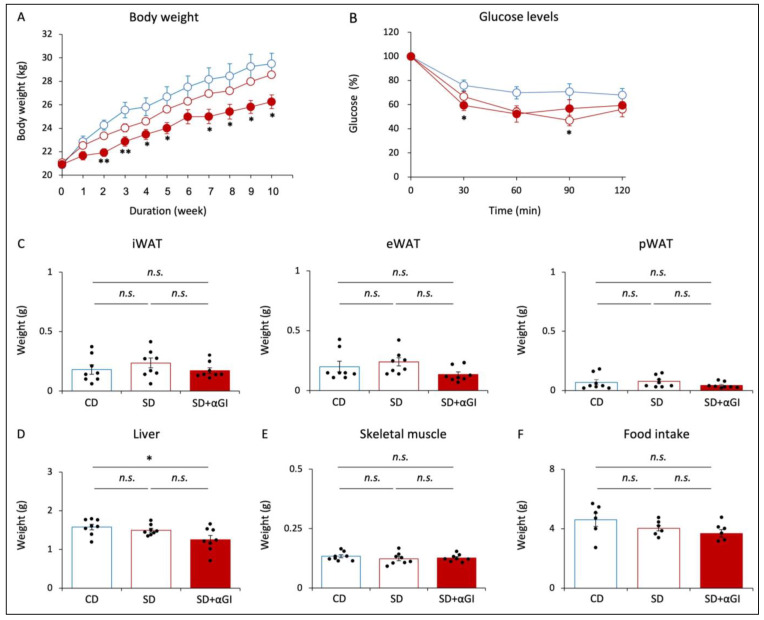
Phenotype of CD mice, SD mice, and SD+αGI mice. CD mice are represented by white circles with blue frames and blue boxes. SD mice are represented by white circles with red frames and red boxes. SD+αGI mice are represented by red circles and red bars. (**A**) Body weight (n = 6). (**B**) Insulin tolerance test (n = 6). (**C**) Weight of inguinal white adipose tissue (iWAT), epididymal WAT (eWAT), perirenal WAT (pWAT), (**D**) liver, and (**E**) skeletal muscle (n = 8). (**F**) Food intake for 24 h (n = 6). * *p* < 0.05, ** *p* < 0.01 vs. CD. *n.s.*: not significant.

**Figure 2 nutrients-16-00196-f002:**
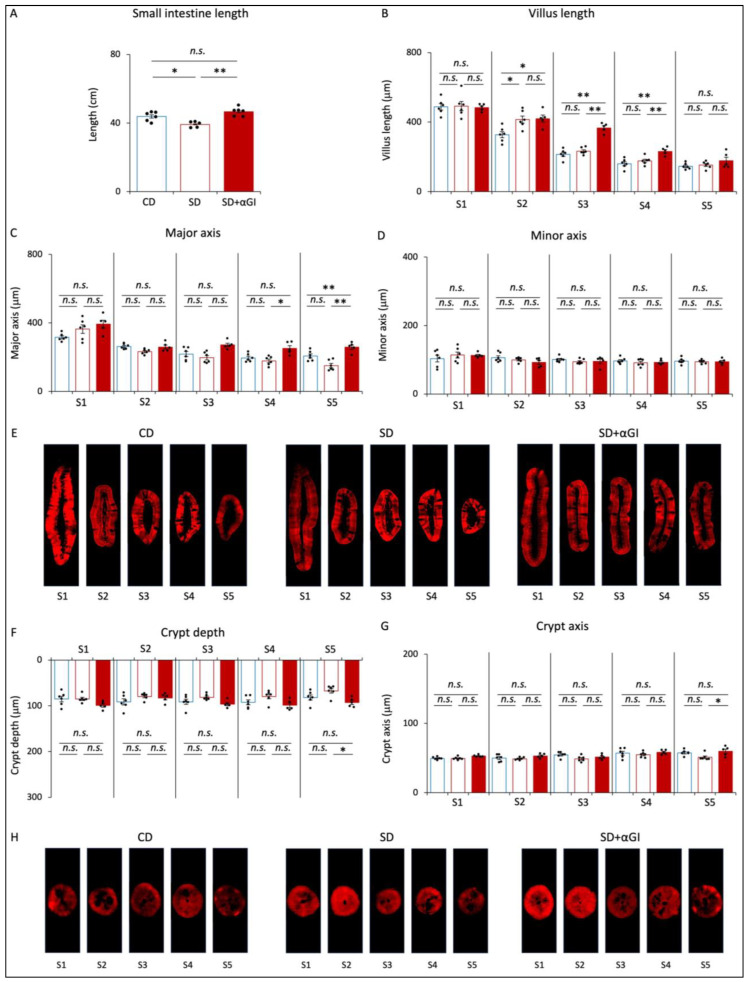
Morphology of the small intestine. CD mice are represented by blue boxes. SD mice are represented by red boxes. SD+αGI mice are represented by red bars. (**A**) Length of small intestine (n = 5–6). (**B**) Villus length. Width of (**C**) villus major and (**D**) villus minor axis, (**F**) crypt depth, and (**G**) crypt axis. The cross section of (**E**) villi and (**H**) crypts (n = 5–6). * *p* < 0.05, ** *p* < 0.01. *n.s.*: not significant.

**Figure 3 nutrients-16-00196-f003:**
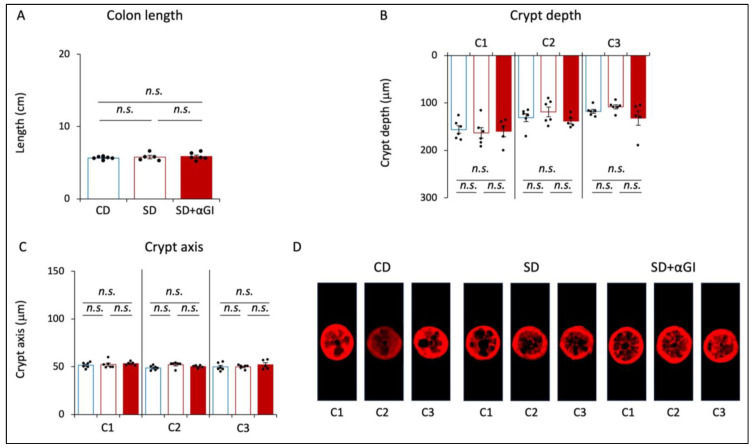
Morphology of the colon. CD mice are represented by blue boxes. SD mice are represented by red boxes. SD+αGI mice are represented by red bars. (**A**) Length of colon, (**B**) crypt depth, and (**C**) crypt axis. The cross section of (**D**) crypts (n = 5–6). *n.s.*: not significant.

**Figure 4 nutrients-16-00196-f004:**
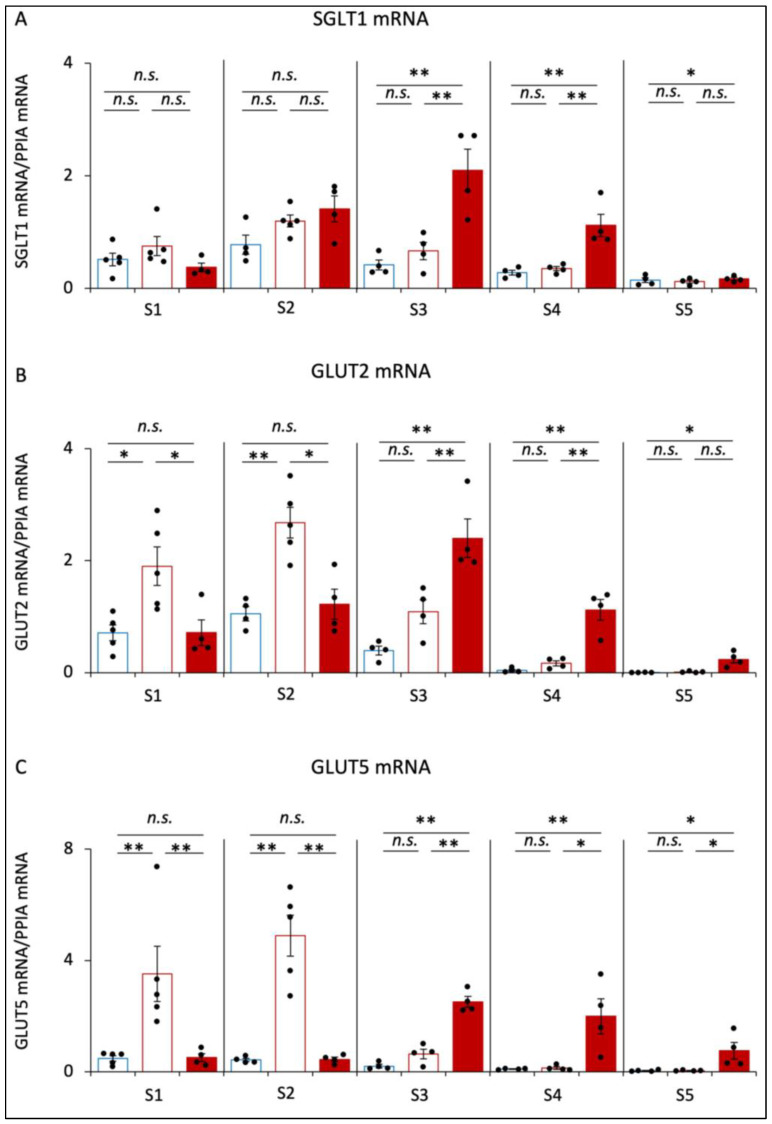
The mRNA expression of glucose and fructose transporters. CD mice are represented by blue boxes. SD mice are represented by red boxes. SD+αGI mice are represented by red bars. (**A**) SGLT1, (**B**) GLUT2, and (**C**) GLUT5 mRNA in small intestine (n = 4–5). * *p* < 0.05, ** *p* < 0.01. *n.s.*: not significant.

**Figure 5 nutrients-16-00196-f005:**
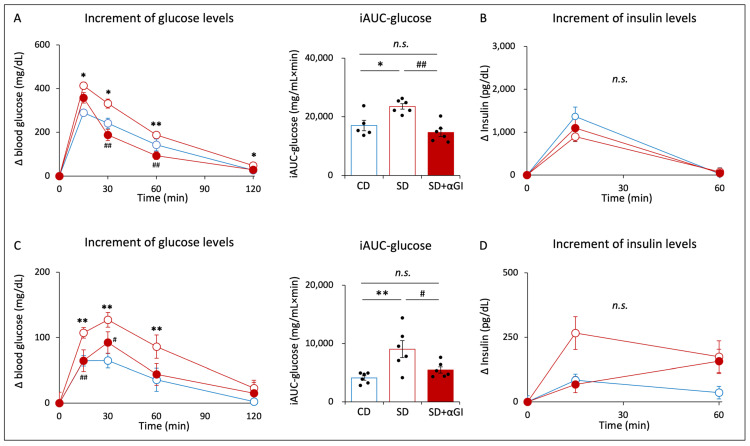
Oral glucose tolerance test (OGTT) and oral fructose tolerance test (OFTT). CD mice are represented by white circles with blue frames and blue boxes. SD mice are represented by white circles with red frames and red boxes. SD+αGI mice are represented by red circles and red bars. iAUC indicates incremental area under the curve. (**A**) The increment in blood glucose levels and (**B**) the increment in insulin levels during OGTT (n = 5–6). (**C**) The increment in blood glucose levels and (**D**) the increment in insulin levels during OFTT (n = 6). * *p* < 0.05, ** *p* < 0.01 vs. CD. ^#^ *p* < 0.05, ^##^ *p* < 0.01 vs. SD. *n.s.*: not significant.

## Data Availability

Data are contained within the article and Appendix A.

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
