# Peer review of "Intestinal Morphology and Glucose Transporter Gene Expression under a Chronic Intake of High Sucrose"

_nutrients, 2024, doi:10.3390/nu16020196_

Round 1
Reviewer 1 Report
Comments and Suggestions for Authors
Summary: This manuscript describes a study that examines the impact of a high-sucrose diet on intestinal physiology, employing advanced 3-D imaging techniques to comprehensively evaluate morphological alterations and gauge the expression levels of sugar transport-associated genes throughout the entire intestine. The study reveals that high sucrose consumption resulted in an upregulation of GLUT2 and GLUT5 mRNA expression, but without inducing significant morphological changes in the intestinal structure. The study has novel aspects such as the segmental and longitudinal determination of mRNA expression and inhibition of sucrose breakdown which simulates delayed exposure of the intestine to the constituent monosaccharides.
Comments:
Abstract:
use of the term "lower" and “upper” when describing the small intestine could be confusing. Please use region (duodenum, jejunum or ilium) or proximal and distal.
Methods:
How were mice housed? Group or individual? Why the 14/10 L/D cycle and not a 12/12?
More detail for ITT – was glucose given during the ITT or was it a simple departure from baseline, fasted glucose only? For the OGTT and OFTT were the mice anesthetized or restrained during the 120 mins?
Intestinal sections – the mouse small intestine is longer than 5 cm. Also, your diagram does not explain how you handled the cecum versus colon. Please clarify this method.
Statistical analysis – please clarify your statistical analysis. Include what tests were performed on the specific data or outcomes. The wording is also not grammatically correct. Please edit with more detail and clarity.
More detailed diet composition should be provided in the diet composition table in the supplementary data. Please include a full ingredients list.
Results:
Title on 3.4: molecules or proteins?
Where are the ITT data? Why report the methods for ITT but not report the data in the results section?
For the OGTT and OFTT, I would suggest that you use an incremental approach to both the time points and AUC analysis. There are differences at T=0 which should be corrected for in this analysis. Reporting the uncorrected data in the supplementary data would be fine but the primary data should be normalized to T=0 blood glucose concentrations.
Comments on the Quality of English LanguagePlease have someone review for grammar. There are some minor issues with some sentences and phrasing.
Reviewer 2 Report
Comments and Suggestions for Authors
Abstract
1. In this section, as well as in the case of the description under Figure 4 - I propose to change the description "mRNA expression level" to "mRNA level" (possibly still "changes") or to "expression on an mRNA level". Since the expression is a process, the proposed description would more closely reflect the nature of the measurement procedure (qPCR). But this is only a suggestion for consideration, optional.
Introduction
1. Line 44 - maybe it is worth considering adding in this sentence what glucose is converted to? In a sentence earlier, the authors indicate the possibility of converting glucose to fructose within the intestinal epithelial cells.
2. Line 53-54 - please consider whether it might be worth modifying the style of the two sentences a bit, emphasizing that GLUT2 is involved in the absorption of the two monosaccharides mentioned, glucose and fructose. In the reviewer's opinion, the construction of the two sentences could confuse the reader and suggest the supposed existence of two different GLUT2 proteins.
2. Materials and Methods
2.1. Experimental animals and diets
1. Line 71-2 - delay of what? surface absorption, absorption, transport - I propose here to clarify the purpose of miglitol.
2. There is no information in the text about the number n of individuals in the groups.
3. It is worth specifying whether regarding the diets used - the test animals received them in a free-access manner or the animals were fed daily per os - for the mentioned 10-week period? For clarity in reading the text in this section.
4. Line 70-78 - I propose to clarify in this paragraph which type of strain (Wt or Villin-1 reporter) is the description of the division into groups? Because from this description in this paragraph, it is not clear. The rest of the text in this section may suggest that it is the Villin-1 strain but this is not clear from the description of the grouping. So if it is a Villin-1 strain, how does this relate to the WT strain? That is - which group is it in this case?
5. Line 83 - What section of skeletal muscle was taken for measurement, was the whole muscle (from which part of the body) or only a fragment? Please, kindly clarify the information in the section.
2.2. ITT, OGTT, and OFTT
1. It is worth clarifying here how, technically, the 4-hour fasting period was achieved since the animals received the diet (as I understand it in a way that was freely available to them), and more precisely: at what point in the 10-week duration of the diet period?
2.3. Quantitative reverse-transcription polymerase chain reaction (RT-PCR)
1. It is worth specifying in one sentence what method of gene expression quantification was used? was it i.e. delta delta CT method or using cDNA dilution curves?
2. It is worth considering when exactly (after the end of the duration of feeding the animals, which day, at what time of the day? etc.) tissue sections were taken for gene expression analysis. if in the same hour - the reader would have more confidence that the authors thus avoided the potential impact of biorhythm on the expression profile of the studied genes in the tissues.
2.5. Image acquisition and processing
1. Despite the link to the supplement to the publication - in my opinion, it would be good to mention how (briefly, concisely) the measurements were made? what software, and how the visualization/measurement was performed. This is a suggestion to consider.
2.6. Statistical analysis
1. Line 125 - is there a missing space error before "SD" in the expression "mean ± 2SD"?
2. In what cases was the Student's t-test used and when was the ANOVA one with Tukey used?
The discussion is written in a concise manner but in receznetna's opinion in a way that is clear to the reader.
What comes to mind is the lack of results of expression analyses at the protein level and/or data on the potential change in density, and tissue localization of the studied targets.
